# Changes in life expectancy and life span equality during the COVID-19 epidemic in 2020-22 in Japan

Yuta Okada[1], Hiroshi Nishiura[1,2]*

**1** School of Public Health, Graduate School of Medicine, Kyoto University, Kyoto, Japan, **2** Center for Health Security, Graduate School of Medicine, Kyoto University, Kyoto, Japan

* nishiura.hiroshi.5r@kyoto-u.ac.jp

## Abstract

### Objectives

Life expectancy at birth is a demographic measure derived from age-specific mortality rates, reflecting the population's mortality pattern and its changes over time. While life expectancy has been heavily applied to quantify the mortality impact of the COVID-19 pandemic, the present study not only decomposes the changes in life expectancy by age and cause of death but also assesses the shifts in the age pattern of mortality in Japan. With this aim, we first evaluated the relationship between life expectancy gap from 2020−21 and 2021−22 and indicators of COVID-19 epidemic size at prefectural level. We also conducted age- and cause-specific decomposition of life expectancy change. Trends of life span equality from 2000−22 were also evaluated at the national level. Prefectural analysis between 2021−22 life expectancy change and annual per-population COVID-19 cases, person days in intensive care, reported COVID-19 deaths did not reveal significant correlations, which was contrary to our analysis from 2020−21. However, decomposition analysis revealed substantial life expectancy shortening attributable to the over-35-year-old population, and large increases in death causes such as cardiovascular or respiratory disorders along with COVID-19. For the total population in Japan, life span equality, an inverse measure of the dispersion in ages at death (higher values indicate less variation), declined in 2020 but increased in 2021 and 2022 despite the shortening in life expectancy. There were several key findings in this work. First, the discrepancy between life expectancy change and COVID-19 statistics in 2022 may, among other factors, possibly be attributable to the growing ascertainment bias of COVID-19. Second, the increased contribution of cardiovascular disorders to life expectancy shortening is an alarming sign for the future. Third, Life span equality in 2021 and 2022 is likely attributed to increased mortality among the elderly.

**Data availability statement:** The codes and data to reproduce the present study are accessible on GitHub at https://github.com/pk2393/LE_h_JP_2020_2022 and are archived on Zenodo (DOI: https://doi.org/10.5281/zenodo.17275790). We used openly accessible COVID-19 statistics from the website of the Ministry of Health, Labour or Welfare, and life tables and related statistics from the website of National Institute of Population and Social Security Research. S1 Data include the results of our numerical analyses. None of the data used in the present study contained personally identifiable information.

**Funding:** This work was supported by the Core Research for Evolutional Science and Technology (grant number JPMJCR24Q3 to HN), the Japan Agency for Medical Research and Development (grant numbers JP24fk0108685 and JP24fk0108710 to HN), the HU-RIZONT International Research Excellence Program (grant number 2024-1.2.3-HU-RIZONT-2024-00034 to HN), the World Health Organization (to HN), the Strategic International Collaborative Research Program (grant numbers JPMJSC20U3 and JPMJSC2105 to HN), the Secom Science and Technology Foundation (to YO), the Ministry of Health, Labour and Welfare (grant numbers 20CA2024, 21HB1002, 21HA2016, and 23HA2005 to HN), and the Research Institute of Science and Technology for Society (grant number JPMJRS22B4 to HN). The funders had no role in the study design, data collection and analysis, decision to publish, or preparation of the manuscript.

**Competing interests:** The authors have declared that no competing interests exist.

# 1. Introduction

Since the start of the COVID-19 pandemic in Wuhan, China in November 2019, evidence of the pandemic's impact on mortality has accumulated globally, with substantial geographical heterogeneity [1–7]. Global studies suggest that from January 1st, 2020 to December 31st, 2021, excess deaths worldwide were in the range of 14.9–15.9 million, with a large proportion attributed to India and the United States [3,4]. Published studies suggest that the global life expectancy change was −1.6 years from 2019 to 2021, when many countries showed bounce-backs from the shortening in 2020. However, other countries faced sustained shortening into 2021 [4–6].

It is now several years since the emergence of COVID-19, and the evaluation of the mortality impact of the condition has become more difficult for several reasons. One reason is changes in the official COVID-19 statistics, which are provided by public health agencies around the world and reflect epidemic activity. These are now less rigorous than in 2020, because most countries have gradually diminished their effort either to control the spread of COVID-19 or to maintain a meticulous surveillance system. Another reason is the change in the nature of deaths associated with COVID-19 since the introduction of vaccines against the disease in late 2020. The direct mortality impact of COVID-19 has been alleviated by these vaccines, but a substantial proportion of deaths are caused indirectly through complications such as cardiovascular disorders, or by limited access to healthcare services when the healthcare capacity or ambulance system were overwhelmed by the increased case load pressure of COVID-19 [5,8–16]. The ongoing emergence of SARS-CoV-2 variants with a high capability of immune evasion and transmission may have worsened the health impact of COVID-19, but understanding the true burden has remained a challenging task. [17,18]

Direct approaches to estimating the mortality impact of COVID-19 are therefore challenging, including in Japan. There, the epidemic size of COVID-19 was greatest upon the emergence of SARS-CoV-2 Omicron (B.1.1.529) lineage variants. In line with other regions, Japan has experienced considerable mortality impact by COVID-19 in terms of excess mortality and life expectancy shortening that have been seen in published studies [1,3–5,15,19–24]. The updated estimates by the National Institute of Population and Social Security Research suggest that life expectancy at birth has shortened for two consecutive years, from 84.58 years in 2021 to 84.10 in 2022 for the total population. Though the shortening itself is rather marginal compared with other countries [25,26], still, the shortening itself is what that have been rarely observed in Japan. However, it is not clear whether the cause-specific impact of this shortening has changed since 2021. Given that published studies in Japan suggested the contribution of COVID-19, senility, cardiovascular disorders, and neoplastic disorders to the increase in age-standardized mortality up to 2022, regarding life expectancy, it is not clear how the contribution of cardiovascular, respiratory, and neoplastic disorders in 2021 have changed from our preceding report [19]. From a demographic perspective, the change in life span equality during and since the COVID-19 pandemic is also interesting. One measure of life span equality, or, evenness of life span, is the logarithm of the inverse of life table entropy. Global and historical demographic analysis suggests that the trends in life expectancy at birth

and life span equality have been in line with each other [27,28]. However, this might not be the case when the age–mortality structure changes drastically. For example, during the COVID-19 epidemic in Japan, the mortality increase in 2021 contributed substantially to shorter life expectancy [19]; however, details on the contribution of age and death causes on life span equality are not evaluated to date. From this perspective, decomposition of the changes in life span equality or related demographic indicators by age and death cause, as proposed and conducted in published studies [29–32], may help understand the nature of mortality impact caused by COVID-19 in Japan.

To examine the demographic impact of the COVID-19 epidemic in 2022 in Japan, we investigated the relationship between reported COVID-19 burden at the prefectural level and life expectancy. We also decomposed annual life expectancy change from 2019–22 by age groups and major causes of death, and evaluated the lifetime loss by age and life span equality during the COVID-19 epidemic.

## 2. Materials and methods

### 2.1. Epidemiological data

We used the data on both complete and abridged life tables, deaths and exposure-to-risk populations available in the Japanese Mortality Database (JMD), which was available for the whole of Japan and by prefecture [33]. Death counts by cause of death and age group were obtained from the vital statistics published by the Ministry of Health, Labour and Welfare of Japan [34]. In line with our previous study, we categorized major causes of death using the International Statistical Classification of Diseases and Related Health Problems 10th Revision (ICD-10) into the top nine major cause categories (based on death counts by cause in 2022), and aggregated the remainder into a single group, to give a total of ten groups [19,34]. The epidemiological data for COVID-19 were retrieved from the open-access data provided by the Ministry of Health, Labour and Welfare [35].

### 2.2. Calculation of period life table and Arriaga decomposition

The deaths counts by cause of death in Japan are only available up to 100 + age group, Thus, for subsequent use for age- and cause-specific decomposition of life expectancy gaps, we re-calculated abridged period life tables from 2000 to 2022 provided by JMD for the whole of Japan and for all prefectures as described before using the standard life table calculation method and shortened the abridged life tables from JMD up to 100 + age group. [19,36–38] (See S1 Methods for details). The recalculation resulted in a minor gap (<0.05 year) in life expectancy compared with that in the original life table by JMD. (relevant data can be found in S1 Data) Using the recalculated abridged life tables, we conducted stepwise Arriaga decomposition of year-on-year changes in life expectancy by age group and by cause of death, taking the earlier year as the reference following the standard calculation [39]. (See S1 Methods for details)

### 2.3. Life expectancy change and COVID-19 statistics at the prefectural level

Three COVID-19 statistics at prefectural level were used for this analysis: (i) annual number of COVID-19 cases, (ii) annual number of person-days in intensive care because of COVID-19, and (iii) annual number of documented deaths due to COVID-19. Using the natural logarithm of each of the COVID-19 indicators as an explanatory variable, we calculated Pearson's correlation coefficients to summarize the linear associations between COVID-19 statistics and life expectancy changes, and conducted linear regression analysis to predict the year-on-year life expectancy change (from the recalculated abridged life tables as above) as the dependent variable for 2020–21 and 2021–22. We evaluated residual normality (Shapiro-Wilk test) and heteroscedasticity (Breusch-Pagan test), and estimated robust standard errors by HC3 covariance estimator that accounts for the relatively small sample size in our analyses [40,41]. Linearity was assessed by comparing linear and quadratic models using a robust Wald test. We also validated our estimates by additionally performing wild bootstrap inference [42]. These inferences were implemented using "sandwich" and "lmtest" packages in R [43,44]. (technical details on the additional analysis to evaluate the year-to-year differences in slopes are provided in S2 Methods).

## 2.4. Changes in life disparity and life span equality $h$

Following Aburto et al., using the complete life tables provided by JMD, we calculated $h = -\log\left(\overline{H}\right)$, which is a measure of life span that is derived from life table entropy $\overline{H}$ [27,37,45–47]. Life table entropy $\overline{H}(t)$ is a measure of variation, or inequality, in life span at time $t$ that is defined as:

$$\overline{H}(t) = -\frac{\int_0^\infty l(x, t)\ln\left(l(x, t)\right) dx}{\int_0^\infty l(x, t) dx} = \frac{e^\dagger(t)}{e_0(t)},$$

Where $e_0$ is the life expectancy at birth, and $e^\dagger(t) = e^\dagger(0, t)$ is the special case of:

$$e^\dagger(x, t) = -\frac{\int_x^\infty l(a, t)\ln\left(l(a, t)\right) da}{l(x, t)} = \frac{\int_x^\infty d(x, t)e(x, t)dx}{l(x, t)},$$

which is the life disparity, or the life expectancy loss after age "$x$" [48]. Note that both $e^\dagger(t)$ and $e_0(t)$ are counted in years, whereas $\overline{H}(t)$ and $h$ are unitless values. We calculated life span disparity $e^\dagger(t)$ and life span equality $h(t) = -\log\left(\overline{H}(t)\right) = \log\left(e_0(t)\right) - \log\left(e^\dagger(t)\right)$ for the period 2000–2022 for the population of Japan.

We chose life span equality $h = -\log\left(\overline{H}\right)$ over life table entropy $\overline{H}$, because it can be intuitively interpreted as an "equality scale", and also that it can be interpreted as the balance between relative changes of $e_0$ and $e^\dagger$:

$$\Delta h = \Delta\log\left(e_0\right) - \Delta\log\left(e^\dagger\right) \approx \frac{\Delta e_0}{e_0} - \frac{\Delta e^\dagger}{e^\dagger}.$$

For example, if $\Delta\log\left(e_0\right) \approx -0.05$ and $\Delta\log\left(e^\dagger\right) \approx -0.15$ result in $\Delta h = 0.1$, this suggests that the relative change in $e_0$ was larger than the relative change in $e^\dagger$ by a factor of $\exp(0.1) \approx 1.105$ as a result of an overall increase in mortality (as reflected in $e_0$) accompanied by a more concentrated distribution of ages at death (higher life span equality/ lower life span dispersion). From a public health viewpoint, comparing the proportional changes in $e_0$ and $e^\dagger$ is informative because $e^\dagger$ summarizes life-years lost (the average remaining life expectancy at death) that cannot be inferred from $e_0$ alone.

To further evaluate $\Delta h$ in 2020−22 in relation to the COVID-19 pandemic, we applied the numerical decomposition method proposed by Horiuchi et al [29] to

- $\Delta\log\left(e_0\right)$, $\Delta\log\left(e^\dagger\right)$: decomposition by age, 2000−01–2021−22

- $\Delta e^\dagger$, and $\Delta h$.: decomposition by age and cause, 2019−20–2021−22

by assuming continuous linear changes in age- and cause-specific mortality along an interpolation path during the time interval of interest. In addition, we assumed that the fractions of mortality rates attributed to each death cause are constant in each age class windows upon which the cause of death data is reported, because the data on the death counts by cause of death is only available in mostly 5-year age classes. (Details on the decomposition framework and numerical calculations are provided in S3 Methods).

Second, to interpret year-on-year $\Delta h$ by age in relation to changes in mortality, following Aburto et al. [27] we calculated
- $w(x, t)W_h(x, t)$: sensitivity of $h$ to the rate of mortality improvement in age group $x$ and at time $t$
- $a^H$: threshold age of $h$ (and $\overline{H}$), which indicates that mortality in those younger than $a^H$ leads to decrease in $h$, and vice versa for mortality in those older than $a^H$ [27] (See S4 Methods for technical details) Note that $w(x, t)W_h(x, t) > 0$ for $x < a^H$ and $w(x, t)W_h(x, t) < 0$ for $x > a^H$, so an increase in mortality (decrease in mortality improvement) in $x > a^H$ leads to increase in $h$, whereas increase in mortality in $x < a^H$ leads to decrease in $h$ These results were compared with year-on-year mortality improvement, i.e., $r(x, t) = \log\left(\mu(x, t)\right) - \log\left(\mu(x, t+1)\right)$, which is analogous to the rate of mortality improvement.

## 2.5. Software

All analyses used R version 4.2.2. [49].

## 2.6. Ethical approval statement

Ethical approval was not required because all data used in the present study did not include any personally identifiable information.

## 3. Results

Our previous work that evaluated the life expectancy in Japan up to 2021, the relationship between life expectancy and epidemiological indicators of COVID-19 by prefecture, and the decomposition of life expectancy change by age and death cause during the earlier period of COVID-19 pandemic. [19] In the following sections, regarding the abovementioned findings in our previous work, we present the update up to 2022, the year when the emergence of Omicron variants has drastically changed the scale of the epidemic and caused substantial strain to the healthcare system in Japan. Beyond extending an array of life expectancy evaluation in our previous work up to 2022, for the same period we also present novel analysis on the change in demographic indicators such as life span equality or life span disparity together with their decomposition by age and death causes.

### 3.1. Life expectancy changes and their correlation with epidemiological indicators of COVID-19

The summary of life expectancies at birth in Japan for the total, male, and female populations from 2019−2022 is shown in Table 1. (Results based on abridged life tables that we re-calculated for use in Arriaga decomposition, which are almost identical to results provided by JMD) Life expectancy of the total population decreased by 0.49 years, from 84.60 to 84.11 from 2021−22. Though the change in the trend of life expectancy was already seen from 2020−21 by 0.15 years (from 84.75 to 84.60 years) as reported in JMD and in our previous work [19], the magnitude of shortening was greater in 2021−22. When compared with the counterfactual life expectancy that is calculated from the 10-year average change in life expectancy up to 2019, the gap was even more prominent. The shortening of life expectancy at birth for male and female populations also grew greater from 2021−22, estimated at 0.42 years (from 81.49 to 81.07 years) and 0.50 years (from 87.63 to 87.13 years), respectively.

Fig 1 shows life expectancy changes of the total population by prefecture in 2019–20, 2020−21, and 2021–2022. Following the drastic change from the overall increasing trend in 2019−20 to the sharply decreasing trend in 2020−21 that was already reported in our previous work [19], all except one prefecture saw a decline in life expectancies from 2021−22. In 2022, the greatest decrease in life expectancy was seen in Iwate (1.00 years), and the only prefecture that enjoyed the increasing trend was Nagasaki (0.05 years). The prefecture level life expectancy changes of male and female population were mostly in line with that of the total population. (For the details, see S1 Data).

**Table 1. Life expectancy of total, male, and female populations in Japan, 2019–22.**

| Population | Total | | Male | | Female | |
|---|---|---|---|---|---|---|
| | Actual | CF 10Y# | Actual | CF 10Y# | Actual | CF 10Y# |
| 2019 | 84.51 | – | 81.42 | – | 87.49 | – |
| 2020 | 84.75 | 84.68 | 81.62 | 81.63 | 87.78 | 87.63 |
| 2021 | 84.60 | 84.84 | 81.49 | 81.83 | 87.63 | 87.76 |
| 2022 | 84.11 | 85.01 | 81.07 | 82.04 | 87.13 | 87.89 |

# Counterfactual life expectancy assuming the same change rate as the average from 2010-2019 up to 2022.

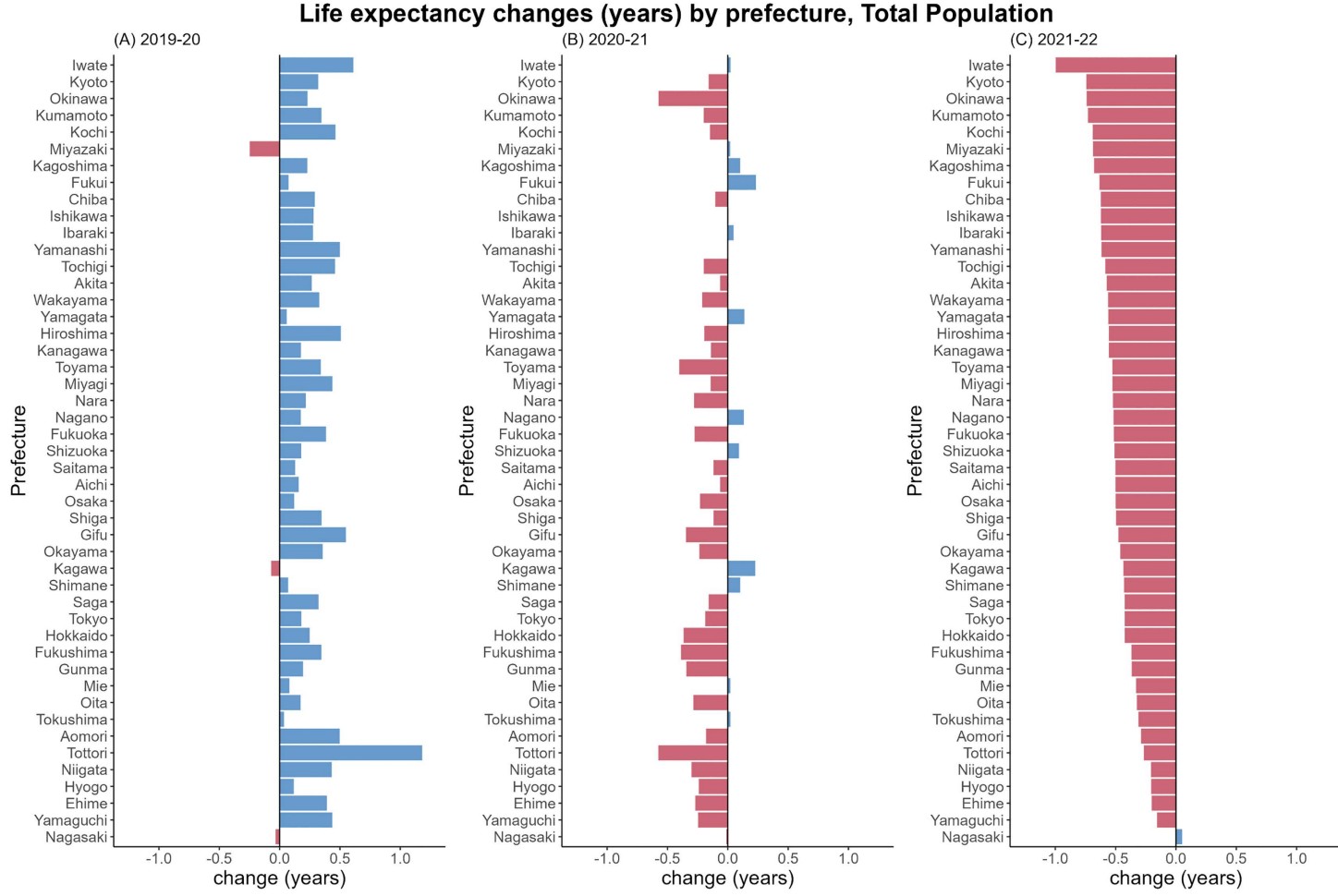

**Fig 1. Life expectancy changes from 2019−20, 2020−21, and 2021−22 by prefecture.** Changes from (A) 2019−20, (B) 2020−21, and (C) 2021−22 are shown. In each panel, bars in blue show positive changes, whereas red bars show negative changes. Order of prefectures are in an ascending order based on the life expectancy change of 2021−22.

Fig 2 presents the association between the reported COVID-19 burden and life expectancy changes at the prefectural level. Pearson's correlation coefficients indicated moderate inverse correlations for cases and person-day under intensive care, and a weaker inverse correlation for deaths due to COVID-19 in 2020−21; correlations were weaker in 2021−22 across all three COVID-19 statistics. In Table 2, the corresponding linear regression results are summarized, with point estimates and 95% confidence intervals computed from robust standard errors. These results were consistent with the patterns observed in Pearson's correlations. Our supplementary analysis did not support significant year-to-year differences in slopes, and the conclusions were unchanged in sensitivity analyses using wild bootstrap inference. Robust Wald tests also supported linear specification over a quadratic model. (S1-S4 Tables).

### 3.2. Arriaga decomposition of life expectancy by age groups and death causes

Fig 3 shows the results of Arriaga decomposition of life expectancy change by age groups and major causes of death. (Aggregated summary by age groups or by causes of death are available as S1 Fig and S2 Fig) As for the contribution

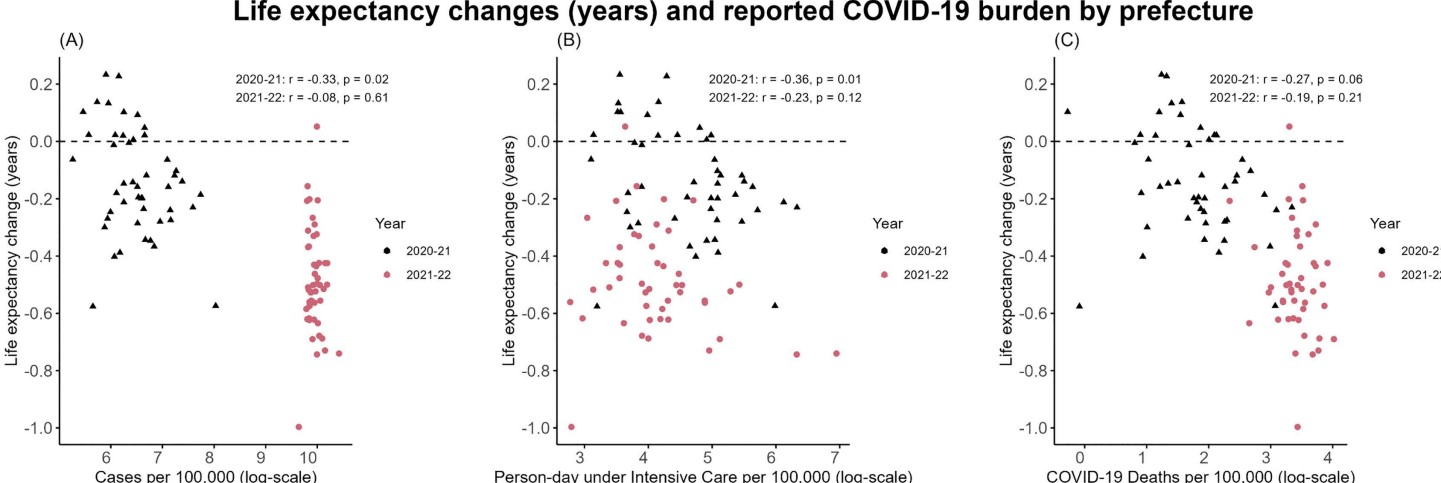

**Fig 2. Correlation between life expectancy changes and COVID-19 burden based on official statistics.** Correlation between life expectancy changes and the reported numbers of (A) annual COVID-19 cases, (B) person-day under intensive care due to COVID-19, and (C) deaths due to COVID-19 are shown. The variables on the x-axis are log-scaled in all panels. In each panel, individual prefectures are presented by black triangles representing 2020−21 data, or red dots representing 2021−22 data, respectively. The horizontal dashed line corresponds to "no year-on-year life expectancy change". On each panel, Pearson's correlation coefficients are overlaid.

**Table 2. Life expectancy changes from 2020 to 2021 and from 2021 to 2022 in relation to COVID-19 statistics: summary of linear regression analysis.**

| COVID-19 data (log-scale) | Period | Coefficient (95% CI*) | Intercept (95% CI) |
|---|---|---|---|
| Cases | 2020−21 | −0.104 (−0.204, −0.003) | 0.534 (−0.137, 1.204) |
| | 2021−22 | −0.103 (−0.696, 0.490) | 0.534 (−5.383, 6.451) |
| Person-days in intensive care | 2020−21 | −0.082 (−0.156, −0.009) | 0.239 (−0.127, 0.604) |
| | 2021−22 | −0.052 (−0.127, 0.023) | −0.272 (−0.609, 0.065) |
| Death | 2020−21 | −0.067 (−0.176, 0.042) | −0.020 (−0.245, 0.205) |
| | 2021−22 | −0.107 (−0.266, 0.053) | −0.125 (−0.675, 0.425) |

*CI: Confidence Interval

by age, the negative contribution among the elderly population in 2020–21 was already seen. However, the negative contribution of the elderly population is more eminent in the change from 2021 to 2022 than in the change from 2020 to 2021. Also, the range of elder age with negative contribution has widened to younger age groups in 2021–22 to as low as 30–34-year-olds.

As for contributions of major death causes by age groups which is also shown in Fig 3, negative contribution by COVID-19 among the elderly population has enlarged substantially in 2021−22, and the total contribution by all ages has grown from −0.096 years in 2020−21 to −0.132 years in 2021−22. In addition to COVID-19, the negative contribution of cardiovascular causes also grew much in 2021−22 especially among age groups older than 50 years old. The total contribution of cardiovascular death was −0.090 years in 2022 and has decreased consistently and substantially compared with +0.070 years in 2020 and −0.005 years in 2021. The negative contribution of "other" causes (the remainder of death causes that do not belong to the top 9 major death cause categories) also increased substantially in 2021−22 among elderly population older than 50 years old, with a total of −0.140 years across all age groups. As in S5 Table, the

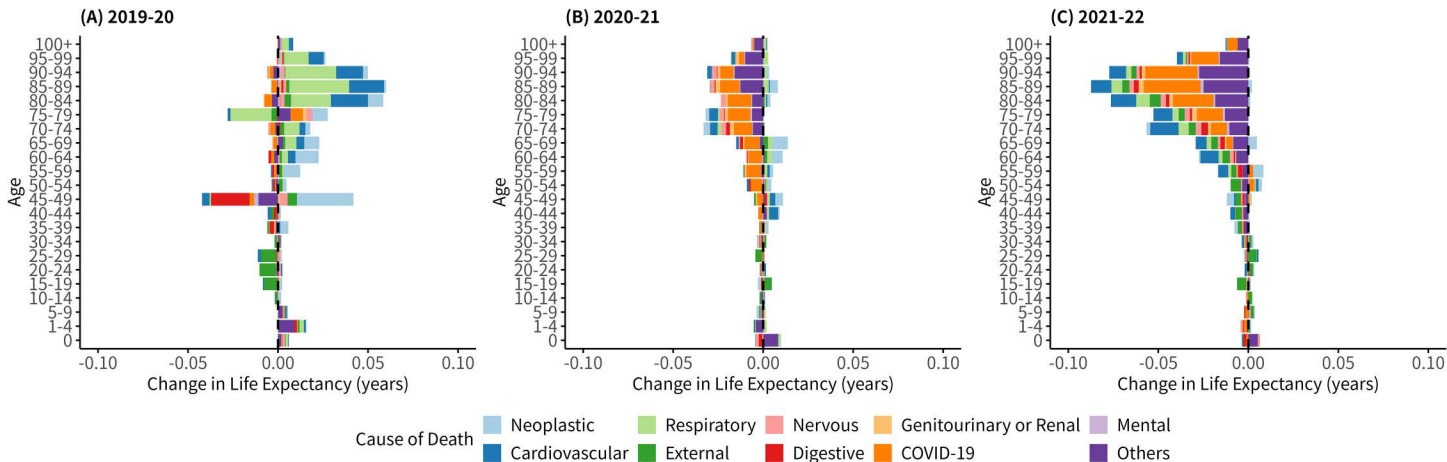

**Fig 3. Arriaga decomposition of life expectancy change by major cause of death and age group, for the entire (total) population of Japan.** Decomposed contribution by age for (A) 2019−20, (B) 2020−21, (C) 2021−22 are shown in each panel. Bar for each major cause is colored as shown in the panel below the plots. Bars representing major causes with positive contribution to life expectancy are stacked on the right-hand side, whereas those with negative contributions are stacked on the left-hand side.

contribution of other causes is likely attributable to "senility" among causes in "other cause" category. A shift toward more negative contributions from 2020−21 and 2021−22 was also observed for respiratory disorders, neoplastic disorders, and other causes. (See S1 Data for detailed results). Results from the decomposition analysis for the male and female populations were similar to that of the total population. (S1Data, S3 Fig and S4 Fig).

### 3.3. Changes in Life span equality and life disparity from 2020−22 in Japan

The values of $h(t)$ as an indicator of life span equality for the total population from 2000 to 2022 are shown in Fig 4. As shown in panel (A), $h$ has mostly increased monotonously up to 2019, with the exception of 2011 when an exceptional number of casualties occurred due to the earthquake and tsunami that hit eastern Japan. That increasing trend was halted in 2020 when the COVID-19 pandemic started, but has resumed to increase since 2021. The values of $h(t)$ for female and male populations also showed very similar patterns to those for the total population. (S5 Fig and S6 Fig) Panel (B) in Fig 4 shows the relationship between $\Delta\log(e_0)$ and $\Delta\log(e^\dagger)$ from 2000 to 2022. Except for outliers 2010−11 and 2011−12 related to the 2011 earthquake, it can be seen that

- In 2019−20 the highest (and positive) $\Delta\log(e^\dagger)$ was observed, which outweighed the positive $\Delta\log(e_0)$ and led to decrease in $h$. ($\Delta e^\dagger$ = 0.095)

- In 2020−21, both $\Delta\log(e_0)$ and $\Delta\log(e^\dagger)$ were slightly negative but $h$ increased as a whole, resulting from relatively larger absolute change in $\Delta\log(e^\dagger)$. ($\Delta e^\dagger$ = −0.064)

- The trend from 2019−20 to 2020−21 went even further and in 2021−22 the greatest negative value of $\Delta\log(e_0)$ was observed. However, it was offset by negative $\Delta\log(e^\dagger)$ and $h$ increased in total. ($\Delta e^\dagger$ = −0.094)

The plots showing the relationship between $h$ and $e_0$ from 2000−22 are provided as S7 Fig, where, in contrast to the long-term positive correlation, decreases in $h$ was seen in 2020 for the first time since 2011, and it was followed by an increase in 2021 and 2022 decreased despite the shortening of life expectancy at birth.

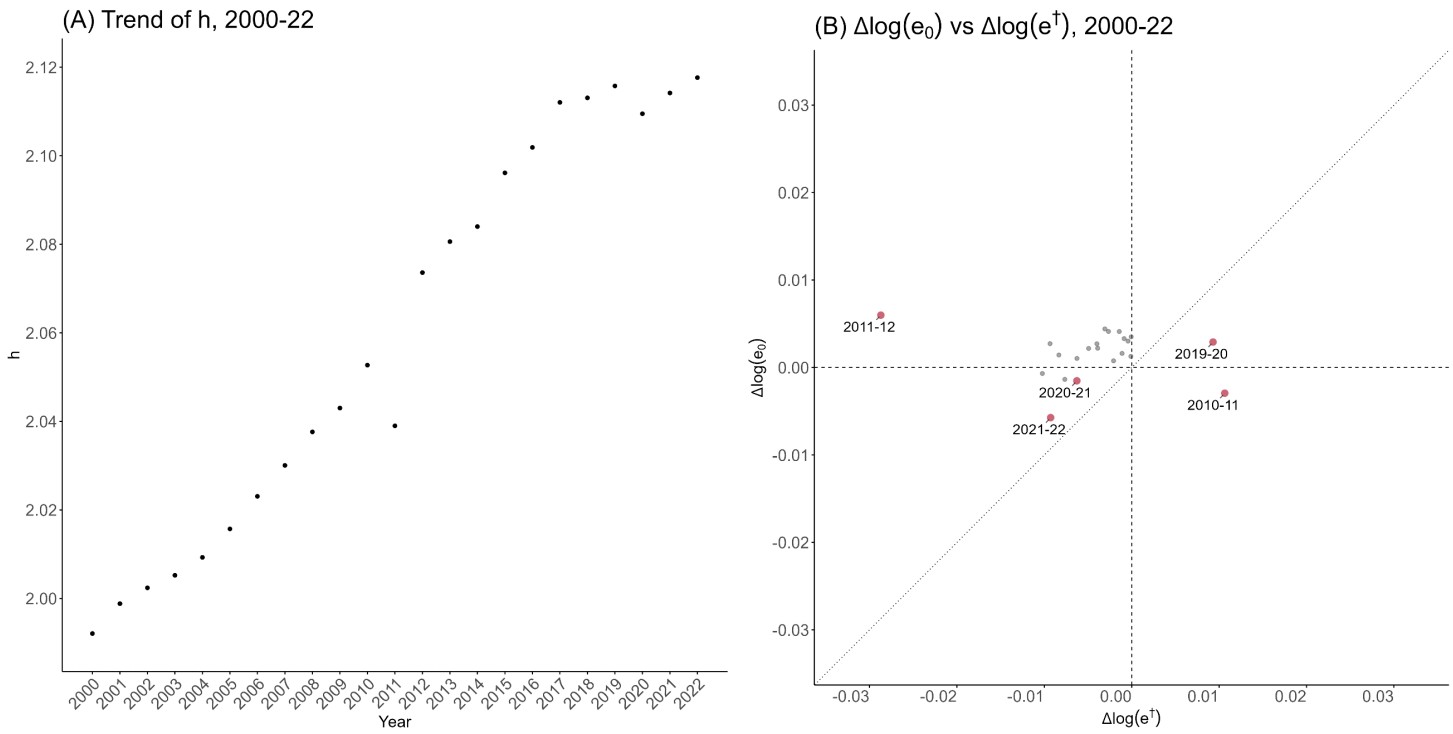

**Fig 4. The trend of life span equality and the underlying dynamics of** $\Delta\log(e_0)$ **and** $\Delta\log(e^\dagger)$ **from 2000 to 2022, for the entire (total) population of Japan.** Panel (A) shows the dynamics of life span equality along time from 2000 to 2022. Panel (B) shows the relationship between the year-on-year difference $\Delta\log(e_0)$ and $\Delta\log(e^\dagger)$ during the same period, where the years corresponding to the red dots are noted within the figure. *Ex: $\Delta h = 0.1$ suggests that the relative change in $e_0$ $\left(\approx \frac{\Delta e_0}{e_0}\right)$ was larger than that in $e^\dagger$ $\left(\approx \frac{\Delta e^\dagger}{e^\dagger}\right)$ by a factor of $\exp(0.1) \approx 1.105$.

Fig 5 shows age- and cause-specific contributions to annual changes in $h$ for 2019−20, 2020−21, and 2021−22 as heatmaps. In 2019−20 ($\Delta h < 0$), negative contributions were concentrated among those older than 85, primarily attributable to cardiovascular and respiratory causes and with smaller negative contributions from external causes in young-adult ages. In 2020−21 ($\Delta h > 0$), the contribution pattern changed drastically; negative contribution from cardiovascular and respiratory causes shrunk in magnitude, while COVID-19 emerged with modest negative contributions mainly at ages 35−84. In contrast, contributions at ages over 85 were positive notably for "other causes". In 2021−22 ($\Delta h > 0$), positive contributions at ages over 85 from COVID-19 and other causes became further strong, whereas the negative contributions at younger age groups for cardiovascular, external, COVID-19, and other causes were also observed. the remaining contributions from other categories were comparatively small with mixed patterns. Same analysis by sex revealed similar patterns. (S8-S9 Fig) The decomposition of life disparity $e^\dagger$ by age and cause are also shown in S10-S12 Fig. In 2021−22, total life disparity decreased by 0.093 years, which was largely driven by 85+ population from an age-perspective. From a cause-of death perspective, COVID-19 and other (remaining) causes of death mostly led this decrease.

Regarding the contribution of mortality rate changes by age to the dynamics of $h(t)$ from 2020 to 2022, we calculated the curves of $w(x, t)W_h(x, t)$ across ages for 2021 and 2022, and also evaluated the year-on-year mortality improvement $r(x, t)$ from 2020 to 2022 for the total population. (S13 Fig) The curves of $w(x, t)W_h(x, t)$ for 2021 and 2022 were very similar, although a slight shift toward the younger ages can be observed in the negative part of the curve in the elderly

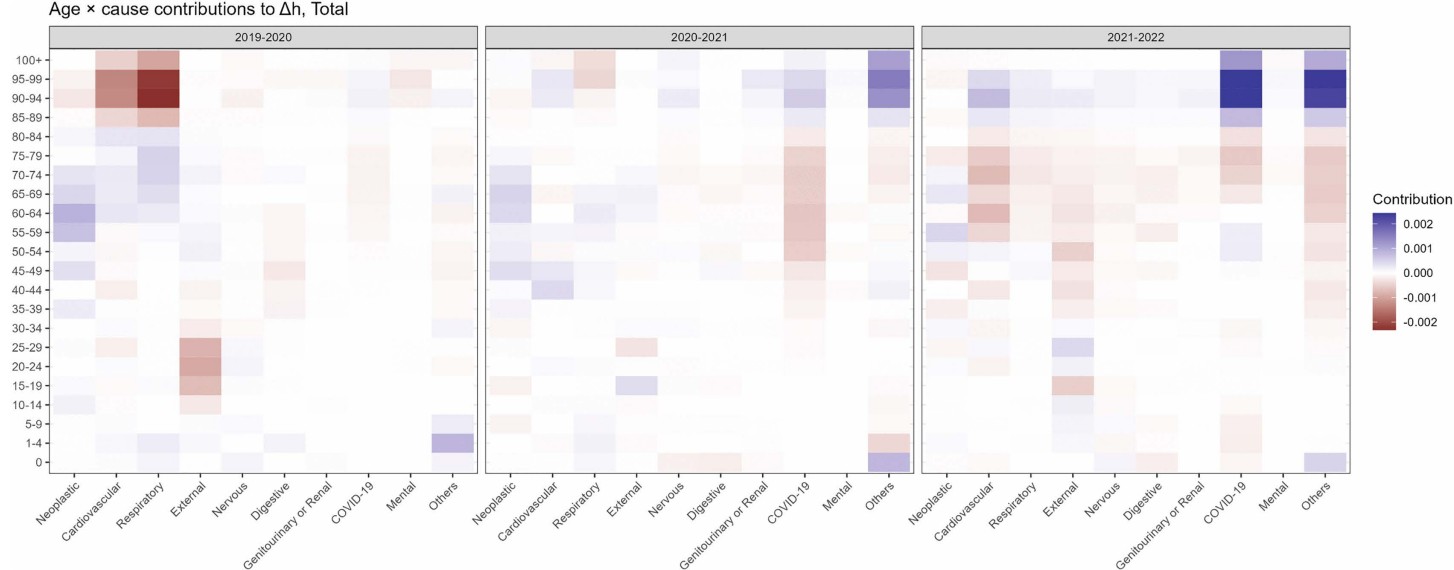

**Fig 5. Age-cause specific contributions to year-on-year changes in $h$ for the entire (total) population of Japan.** In each panel for 2019−20 (Left), 2020−21 (Middle), and 2021−22 (Right), positive and negative contributions to $\Delta h$ are represented by blue and red, respectively, with gradations in color that express the magnitude of contributions.

population. As for $r(x, t)$, $r(x, 2020)$ above $x = a^H$ lay in the positive range, whereas $r(x, 2021)$ and $r(x, 2022)$ mostly lay in the negative range for the age range. For ages younger than $x = a^H$, the signs of $r(x, 2020)$, $r(x, 2021)$, and $r(x, 2022)$ were inconsistent across different ages, suggesting that increased mortality among ages older than $a^H$ have clearly contributed to the increase of $h(t)$ in 2021 and 2022. (Also see S14 Fig and S15 Fig for results on female and male populations). This is in line with the opposite contributions to changes in $h$ above or below roughly 80 years of age for each cause of death that was observed in Fig 5.

## 4. Discussion

Our study showed the pattern of deaths in Japan during the COVID-19 epidemic (up to 2022) through demographic information. The main finding was the growing impact of the older population and cardiovascular deaths on the shortening of life expectancy, which was considerable from 2021 to 2022. The unclear correlations across prefectures between life expectancy change and epidemiological indicators of the COVID-19 burden from 2022 is also a concern, as it is possibly linked to the low detection of COVID-19 cases and associated deaths, among other potential factors such as strain in healthcare capacity or surges in other infectious diseases such as influenza. The increasing trend in life span equality despite the life expectancy shortening turned out to be a result of substantial increase in mortality among the older population, which is likely attributed to death caused by COVID-19 or "other (remaining) causes" (mostly due to senility). While Japan has experienced relatively small mortality losses during the COVID-19 pandemic from a global comparison perspective, the abovementioned findings highlight the importance of monitoring the pandemic impact through a demographic lens. [3,7]

There were two key findings from our study. The first was that all age groups over 30 years old contributed to the shortening of life expectancy in 2022, as shown in Fig 3 and S1 Fig. However, compared with the overall shortening attributed to age groups over 50 in 2021, the negative impact was more diffuse across ages. This finding is similar to what was

observed in 2020–21 in countries in Eastern Europe, though the underlying situations in these countries, such as types of circulating SARS-CoV-2 variants, vaccine coverage, and healthcare situations, would have been quite different from that in Japan from 2021–22. [12–14,47] In Japan, the population-wide vaccine coverage of the second dose of mRNA vaccines (BNT162b2 [Pfizer/BioNTech] and mRNA-1273 [Moderna] vaccines) was around 80% by the end of 2021, and the coverage of the third dose also increased from around 15% at the end of 2021 to 68% by the end of 2022 [50]. Despite this high vaccination rate, we found substantial mortality caused by COVID-19 in Japan among wider age groups in 2022. It is possible that this was not fully captured by COVID-19 statistics, as suggested in our prefectural analyses (Fig 2 and Table 2). Provided that not only the testing and reporting practices but also the healthcare-seeking behaviour among citizens have evolved over the course of time, our findings support the need for complementary monitoring based on vital statistics, including timely all-cause and cause-specific mortality surveillance.

Another key finding was the substantial growth in the negative contribution of cardiovascular disorders to life expectancy shortening, especially among populations over 50 years old (Fig 3 and S2 Fig). This was not surprising, because several published studies have shown an elevated risk of cardiovascular diseases associated with COVID-19 [13–15,51–53], which resulted in impacts that contribute to shortening of life expectancy across various countries [54]. Based on these literatures, it is likely that the magnitude of life expectancy shortening in Japan caused by cardiovascular deaths in 2022 aligns with the global trend during the COVID-19 pandemic. Though there is no publicly available death certificate data with multiple-cause-of-death information, given a published study in Japan suggesting a similar trend in both the increase in age-standardized mortality rates in Japan from 2020−21–2021−22 caused by COVID-19 and heart disease [24], it is possible that a fraction of cardiovascular deaths was directly attributable to COVID-19.

About cause-specific contributions to life expectancy change other than cardiovascular disorders, the negative change in contributions by respiratory causes from 2021 to 2022 and the consistently negative trend in contributions by neoplastic disorders since 2020 are also of note. In addition to COVID-19-associated conditions, these findings may be attributable to an array of factors including changes in hospital attendance [19]. Because there is a gap between this finding and the global and regional cause-specific contributions to life expectancy change from 2019–21 [55], further update on this issue is warranted to evaluate changes in life expectancy change by causes of death. Another finding is the increase in the contribution of remaining causes of death, which is mostly explained by the sharp increase in deaths due to senility after 2021 (S5 Table) [34]. Overall, our findings on the relationship between cause of death and life expectancy change suggests that the trend change in mortality by cause of death in Japan in 2021 which was reported in previous studies became more prominent in 2022 [19,23,24]. Together with preceding studies in Japan, our findings highlight the importance of strengthening health-system resilience to maintain capacity not only for COVID-19-associated illnesses but also for non-COVID critical illnesses even during pandemic surges.

The changes in life span equality $h$ during the COVID-19 pandemic were also of note. Our result from age-cause decomposition of $\Delta h$ highlights that the overall decrease in $h$ in 2019−20 was not only led by cardiovascular and respiratory causes in elder population but also by external causes in young-adult ages. As shown in S6 Table (and in S1 Data), our supplementary analysis shows that the increase in mortality among young-adults aged 10−44 that were led by external causes was essentially attributable to suicide, which aligns with previous reports on the increase of suicide during 2020 in Japan [56–58]. In contrast, the overall increase in $h$ after 2020−21 is largely attributable to mortality in those older than 80 by COVID-19 and "other" causes of death. Together with our results from Arriaga decomposition, this change in life span equality $h$ from 2020−22 can actually be interpreted as an consequence of elevation in mortality that heavily affected the elderly population, that resulted in the shortening of life expectancy at birth. These findings add to demographic case studies on the historical relationship between life expectancy and life span equality [27,28].

Our study had some limitations. First, we could not examine the relationship between COVID-19 and other causes of death in detail at the prefectural level, because data on prefectural death count stratified by age and cause of death are not openly accessible. Detailed analysis of prefectural data would have provided insights on geographic heterogeneity,

and we hope to explore this in the future. Second, we ignored geographic and temporal variation in the ascertainment bias for COVID-19 statistics. We sufficiently met our key focus to be confident about the true mortality burden of COVID-19, but these factors could have biased our analysis of the relationship between prefectural COVID-19 statistics and life expectancy change. Third, our analyses based on cause of death data from the vital statistics did not consider multiple-cause-of-death information, which is not publicly available. Thus, for example, some deaths that we classified as cardiovascular deaths may have involved COVID-19 as an immediate, intermediate, or contributory cause, and vice versa. This potential misclassification could be evaluated if we gain access to the individual-level death certificate data that list all the mentioned causes. Fourth, we did not consider the fluctuation in the coverage of death registrations in Japan from 2019–22. However, it is unlikely that we missed a large proportion of deaths that would substantially affect our results, because the completeness of death registration is reported to be 90–99% in Japan [59].

In conclusion, our demographic analysis showed the impact of the COVID-19 epidemic up to 2022, when the epidemic grew substantially larger. The demographic burden of the pandemic increased more in 2022 than in 2021 or before, but the COVID-19 burden reported by epidemiological surveillance may not have captured this trend, that was suggested in our prefectural analysis and the increasing share of deaths coded as senility as a contributing factor to life expectancy shortening. This is probably due to both the shrinking coverage of epidemiological surveillance and the growing impact of COVID-19-associated deaths caused by complications such as cardiovascular disorders. We also showed that the increase in life span equality after 2020−21 was largely attributable to higher mortality among older people, though the impact of suicide among young adults remains a matter of concern. Our study therefore provides valuable insights into the mortality impact of the COVID-19 epidemic in Japan, and sheds light on important policy implications: incorporating timely demographic analysis by age and cause based on vital statistics into routine epidemiological surveillance, securing access to emergency and time-sensitive healthcare (especially for cardiovascular conditions) during epidemic surges, and implementing public health programs to support mental health and prevent frailty, especially during pandemic situations.

## Supporting information

**S1 Data. Results of demographic analysis in the present study.**
(XLSX)

**S1 Methods. Life table calculation in the present study.**
(DOCX)

**S2 Methods. Statistical details and additional analyses for the prefectural analysis of the relationship between COVID-19 statistics and life expectancy changes.**
(DOCX)

**S3 Methods. Numerical details of the decomposition of changes in life span equality h and related demographic indicators.**
(DOCX)

**S4 Methods. Changes in life span equality and its sensitivity to changes in mortality.**
(DOCX)

**S1 Fig. Arriaga decomposition of life expectancy change by age group for the total population of Japan.** Decomposed contribution by age for (A) 2019–20, (B) 2020–21, (C) 2021–22 are shown in each panel. Blue bars show positive contributions, and red bars negative contributions.
(TIF)

**S2 Fig. Arriaga decomposition of life expectancy change by major causes of death of Japan.** Decomposed contribution by age for (A) 2019–20, (B) 2020–21, (C) 2021–22 are shown in each panel. As in S1 Fig, blue bars show positive contributions, and red bars negative contributions.
(TIF)

**S3 Fig. Arriaga decomposition of life expectancy change by major cause of death and age group, for the female population of Japan.** Decomposed contribution by age for (A) 2019–20, (B) 2020–21, (C) 2021–22 are shown in each panel. The key for the colors of the bars is shown in the panel below the plots. Bars for major causes with positive contributions to life expectancy are stacked on the right-hand side, and those with negative contributions are on the left-hand side.
(TIF)

**S4 Fig. Arriaga decomposition of life expectancy change by major cause of death and age group, for the male population of Japan.** Decomposed contribution by age for (A) 2019–20, (B) 2020–21, (C) 2021–22 are shown in each panel. The key for the bar colors are shown in the panel below the plots. Bars representing major causes with positive contributions to life expectancy are stacked on the right-hand side, and those with negative contributions are on the left-hand side.
(TIF)

**S5 Fig. The trend of life span equality and the underlying dynamics of $\Delta\log\left(e_0\right)$ and $\Delta\log\left(e^\dagger\right)$ from 2000 to 2022, for the female population of Japan.** Panel (A) shows the dynamics of life span equality along time from 2000 to 2022. Panel (B) shows the relationship between the year-on-year difference $\Delta\log\left(e_0\right)$ and $\Delta\log\left(e^\dagger\right)$ during the same period, where the years corresponding to the red dots are noted within the figure.
(TIF)

**S6 Fig. The trend of life span equality and the underlying dynamics of $\Delta\log\left(e_0\right)$ and $\Delta\log\left(e^\dagger\right)$ from 2000 to 2022, for the male population of Japan.** Panel (A) shows the dynamics of life span equality along time from 2000 to 2022. Panel (B) shows the relationship between the year-on-year difference $\Delta\log\left(e_0\right)$ and $\Delta\log\left(e^\dagger\right)$ during the same period, where the years corresponding to the red dots are noted within the figure.
(TIF)

**S7 Fig. The comparison of trend in life expectancy and life span equality from 2000 to 2022.** 2D plots for (A) total, (B) female, and (C) male population are shown. Each panel represents the relationship between life expectancy and life span equality during this period. The years corresponding to the red dots are noted within the figure.
(TIF)

**S8 Fig. Age-cause specific contributions to year-on-year changes in *h* for the female population of Japan.** In each panel for 2019−20 (Left), 2020−21 (Middle), and 2021−22 (Right), positive and negative contributions to $\Delta h$ are represented by blue and red, respectively, with gradations in color that express the magnitude of contributions.
(TIF)

**S9 Fig. Age-cause specific contributions to year-on-year changes in *h* for the male population of Japan.** In each panel for 2019−20 (Left), 2020−21 (Middle), and 2021−22 (Right), positive and negative contributions to $\Delta h$ are represented by blue and red, respectively, with gradations in color that express the magnitude of contributions.
(TIF)

**S10 Fig. Summary of contribution to year-on-year change of $e^\dagger$ by age and cause for the entire (total) population of Japan.** In both panel A and B, each row represents the changes of $e^\dagger$ in 2019−20, 2020−21, and 2021−22 that is decomposed by age or cause.
(TIF)

  

**S11 Fig. Summary of contribution to year-on-year change of $e^\dagger$ by age and cause for the female population of Japan.** In both panel A and B, each row represents the changes of $e^\dagger$ in 2019−20, 2020−21, and 2021−22 that is decomposed by age or cause.
(TIF)

**S12 Fig. Summary of contribution to year-on-year change of $e^\dagger$ by age and cause for the male population of Japan.** In both panel A and B, each row represents the changes of $e^\dagger$ in 2019−20, 2020−21, and 2021−22 that is decomposed by age or cause.
(TIF)

**S13 Fig. Weight of change in life span equality and mortality improvement by age in Japan, total population.** Panel (A) shows the weight $w(x, t)W_h(x, t)$ for $t = 2022$ (solid blue line) and $t = 2021$ (dashed red line). Panel (B) describes the year-on-year mortality improvement $r(x, t) = \log(\mu(a, t)) - \log(\mu(a, t + 1))$ for $t = 2020$ (red), $t = 2021$ (green), and $t = 2022$ (blue). Vertical dashed lines in both panels represent the threshold age $a^H = 84.07$ for 2022.
(TIF)

**S14 Fig. Weight of change in life span equality and mortality improvement by age in Japan, female population.** Panel (A) shows the weight $w(x, t)W_h(x, t)$ for $t = 2022$ (solid blue line) and $t = 2021$ (dashed red line). Panel (B) describes the year-on-year mortality improvement $r(x, t) = \log(\mu(a, t)) - \log(\mu(a, t + 1))$ for $t = 2020$ (red), $t = 2021$ (green), and $t = 2022$ (blue). Vertical dashed lines in both panels represent the threshold age $a^H = 86.99$ for 2022.
(TIF)

**S15 Fig. Weight of change in life span equality and mortality improvement by age, male population.** Panel (A) shows the weight $w(x, t)W_h(x, t)$ for $t = 2022$ (solid blue line) and $t = 2021$ (dashed red line). Panel (B) describes the year-on-year mortality improvement $r(x, t) = \log(\mu(a, t)) - \log(\mu(a, t + 1))$ for $t = 2020$ (red), $t = 2021$ (green), and $t = 2022$ (blue). Vertical dashed lines in both panels represent the threshold age $a^H = 80.97$ for 2022.
(TIF)

**S1 Table. Prefectural linear regression analysis, residual diagnostics (Shapiro–Wilk and Breusch–Pagan tests).**
(DOCX)

**S2 Table. Prefectural linear regression analysis, validation of confidence interval estimates by wild bootstrap method.**
(DOCX)

**S3 Table. Prefectural linear regression analysis, comparison between linear and quadratic model.**
(DOCX)

**S4 Table. Prefectural linear regression analysis, slope change from 2020−21–2021−22.**
(DOCX)

**S5 Table. Trend of death attributable to senility from 2019 to 2022.**
(DOCX)

**S6 Table. Change in deaths per 100k population due to COVID-19, suicide, and remaining causes in those aged 10−44 in 2019−20.**
(DOCX)

## Acknowledgments

We thank Melissa Leffler, MBA from Edanz (https://jp.edanz.com/ac) for editing a draft of this manuscript.

## Author contributions

**Conceptualization:** Yuta Okada, Hiroshi Nishiura.

**Data curation:** Yuta Okada.

**Formal analysis:** Yuta Okada.

**Funding acquisition:** Hiroshi Nishiura.

**Investigation:** Yuta Okada.

**Methodology:** Yuta Okada, Hiroshi Nishiura.

**Supervision:** Hiroshi Nishiura.

**Validation:** Yuta Okada, Hiroshi Nishiura.

**Visualization:** Yuta Okada.

**Writing – original draft:** Yuta Okada, Hiroshi Nishiura.

**Writing – review & editing:** Yuta Okada, Hiroshi Nishiura.

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
