## [Decision Letter · Decision Letter 0]

2 Dec 2025

Dear Dr. Nishiura,

Thank you for submitting your manuscript to PLOS ONE. After careful consideration, we
feel that it has merit but does not fully meet PLOS ONE’s publication criteria as it
currently stands. Therefore, we invite you to submit a revised version of the
manuscript that addresses the points raised during the review process.

Please submit your revised manuscript by Jan 16 2026 11:59PM. If you will need more
time than this to complete your revisions, please reply to this message or contact
the journal office at plosone@plos.org. When
you're ready to submit your revision, log on to https://www.editorialmanager.com/pone/ and select the 'Submissions
Needing Revision' folder to locate your manuscript file.. When you're ready to
submit your revision, log on to https://www.editorialmanager.com/pone/ and select the 'Submissions
Needing Revision' folder to locate your manuscript file.. When you're ready to
submit your revision, log on to https://www.editorialmanager.com/pone/ and select the 'Submissions
Needing Revision' folder to locate your manuscript file.. When you're ready to
submit your revision, log on to https://www.editorialmanager.com/pone/ and select the 'Submissions
Needing Revision' folder to locate your manuscript file.

If you would like to make changes to your financial disclosure, please include your
updated statement in your cover letter. Guidelines for resubmitting your figure
files are available below the reviewer comments at the end of this letter.

We look forward to receiving your revised manuscript.

Kind regards,

Claudio Alberto Dávila-Cervantes, Ph.D.

Academic Editor

PLOS ONE

Journal Requirements:

4. We note that there is identifying data in the Supporting Information file
“supplementaryfiles.zip”. Due to the inclusion of these potentially identifying
data, we have removed this file from your file inventory. Prior to sharing human
research participant data, authors should consult with an ethics committee to ensure
data are shared in accordance with participant consent and all applicable local
laws.

-Location data

Reviewer's Responses to Questions

**Comments to the Author**

1. Is the manuscript technically sound, and do the data support the conclusions?

Reviewer #1: Yes

Reviewer #2: Yes

2. Has the statistical analysis been performed appropriately and rigorously?


Reviewer #1: No

Reviewer #2: Yes

3. Have the authors made all data underlying the findings in their manuscript fully
available?

Reviewer #1: Yes

Reviewer #2: Yes

4. Is the manuscript presented in an intelligible fashion and written in standard
English?

Reviewer #1: No

Reviewer #2: No

Reviewer #1: The study examines the decrease in life expectancy in Japan during the
COVID-19 pandemic. The main focus is on the life expectancy changes in 2021 and
2022. These changes are decomposed by age and major classes of causes of death. A
drop in a life-table measure of interindividual life-span equality in 2020 is
emphasized. In addition to the national-level analyses, there is also an analysis of
associations between the life expectancy changes in 2020-22 and indicators of the
population spread of COVID-19 across prefectures.

Herewith, I comment on shortcomings in the manuscript.

1. The Introduction and Discussion sections do not mention some relevant studies on
the COVID-related mortality elevation in Japan. Namely, these are:

Hirokazu Tanaka, Shuhei Nomura, Kota Katanoda, Changes in Mortality During the
COVID-19 Pandemic in Japan: Descriptive Analysis of National Health Statistics up to
2022, Journal of Epidemiology, 2025, Volume 35, Issue 3, Pages 154-159

Tanaka H, Togawa K, Katanoda K. Impact of the COVID-19 pandemic on mortality trends
in Japan: a reversal in 2021? A descriptive analysis of national mortality data,
1995–2021. BMJ Open 2023;13:e071785. doi:10.1136/ bmjopen-2023-071785

Z Shervani,AA Khan, I Khan, A Sherwani, PDM Kumar et al. Marginal Shortening of Life
Expectancy in Japan During COVID-19: A Low Pandemic Impact Country Due to Improved
Health Infrastructure and Awareness. (2024). European Journal of Medical and Health
Sciences, 6(6), 9-17. https://doi.org/10.24018/ejmed.2024.6.6.2214

Zameer Shervani, Aamir Akbar Khan, Intazam Khan et al. Factors Explaining Japan’s Low
COVID-19 Mortality: Comparison with Rich and Democratic Countries. (2025). European
Journal of Medical and Health Sciences, 7(1), 1-9. https://doi.org/10.24018/ejmed.2025.7.1.2245

Mst Sirajum Munira, Okada Y, Nishiura H. 2023. Life-expectancy changes during the
COVID-19 pandemic from 2019–2021: estimates from Japan, a country with low pandemic
impact. PeerJ 11:e15784 DOI 10.7717/peerj.15784

These recent studies are thematically close to the present study. They should be
mentioned in the Introduction and/or Discussion.

Similarities or differences between the present study and the earlier studies should
be noted. The readers should know what novelty the present study adds to the
existing literature.

2. The study focuses on annual changes in e0 in 2020-22. However, the full amount of
e0 losses in these years can be assessed only through comparisons between the
observed e0 values and the counterfactual (predicted) e0 values. Using the Human
Mortality Database (HMD) series for Japan, one can see that from 2012 to 2019, the
Japanese life expectancy was increasing by 0.18 years per calendar year on average.
If we apply this rate of increase to 2020-22, the e0 losses for Japan in 2021 and
2022 would be 0.38 and 0.98 years instead of the observed (annual declines) of 0.13
and 0.48 years.

This difference is important and should be acknowledged. It would also be interesting
to see speculations about possible differences between the annual decreases in e0 in
2020-22 and the e0 losses in the same years.

3. The manuscript has a massive Methods section. It is not clear whether it is a
methodology paper or a substantive analysis. There are numerous formulae. All of
them are known from earlier studies. The first seven equations are completely
unnecessary, as they are very conventional and can be found in demography textbooks,
such as those by Preston or Keyfitz, or Chiang.

4. The following equations for the analysis of the age at death equality h are less
trivial since they are newer.

Maybe the formal definitions of the entropy H, e+ (e-dagger), and the equality h
could remain in the main text.

The next four equations related to further analysis of temporal changes in h should
be given in a supplementary appendix.

Instead, the reader needs a transparent explanation of all measures and calculation
procedures. The substantive sense of h should be clarified. Without a transparent
explanation, it's impossible to understand why h is used and what is the public
health sense of its decrease by 0.07 between 2019 and 2020.

Perhaps it would be better to use e+ instead of h. e+ is a transparent measure of
life expectancy losses by Vaupel and Canudas-Romo. e+ is counted in years of life
and has a clear public health sense (Shkolnikov et al. 2011).

Shkolnikov V.M., Andreev E.M., Zhang Z., Oeppen J., J.W.Vaupel. 2011. Losses of
expected lifetime in the United States and other developed countries: methods and
empirical analyses. Demography, 48: 211-239

5. Despite the length of the Methods, they lack a precise description of the data and
computational procedures. Which calculations were based on the complete life tables
vs. the abridged life tables? It is clearly preferable to use the single-year-age
data for calculating dispersion measures such as h or e+.

This information is needed. It can be provided in a supplementary appendix.

6. The analysis of causes of the h changes between 2019 and 2020, 2020 and 2021, 2021
and 2022 is not convincing.

From the methodology point of view, it is possible to see directly the components of
changes in h. This could be done by decomposing the change in h by age and CODs with
one of the universal decomposition methods (discrete replacement method by Andreev
et al. 2002; continuous change method by Horiuchi et al. 2008). These methods
estimate additive contributions of different ages and causes of death to a change in
any aggregate measure. This has not been done. Looking at changes in M(x) or at the
weights W(x) cannot replace the direct decomposition.

On the substantive side, there are also some problems.

First, the alarming drop in h between 2019 and 2020 is not explained. However, Figure
3 reveals moderate mortality increases at young adult ages in 2020. My own quick
look at the Human Mortality Database data for Japan also indicates mortality
increases at ages between 20 and 29 as well as some ages between 30 and 44 in 2020.
So, what happened in 2020? It is important to know. The potential role of the
pandemic is doubtful.

Second, increases in h from 2020 to 2021 and from 2021 to 2022 are considered
“undesirable” as they were (likely) caused by the mortality elevation at old and
very old ages.

At the same time, the h increase in 2021-22 may be considered an advantage of Japan
compared to other countries. While in Japan, the mortality increase in 2021 and
especially in 2022 was entirely concentrated at old ages, in many other countries
(USA, Eastern Europe, Britain, Belgium, and other), the COVID-related mortality
increases were also large at midlife and “young old” ages (55-69). The latter
pattern results in stagnant or even decreasing values of h.

That is to say that the changes in e0 and e+ in Japan should be considered in context
of e0 and e+ changes elsewhere.

7. Discussion about the large increases in mortality from CVD in 2021 and 2022 does
not take into account potential direct contributions of COVID-19 to deaths from CVD.
COVID-19 could appear on the medical death certificate as an immediate,
intermediate, or contributory cause.

It would be good to discuss how large the direct influence of COVID-19 could be in
CVD deaths in Japan.

8. The analysis of statistical associations between e0 decreases in 2021 and 2022 and
the epidemiological indicators is not fully justified.

The authors should perform statistical tests for normality, heteroscedasticity, and
linearity.

Pearson’s r values together with p-values should shown on these graphs.

The regression outcomes are commented as if there is a radical difference between
2021 and 2022. It looks like statistical links exist in 2021 and they do not exist
in 2022. However, the scatterplots and the tables show that in both years, the links
are very weak, but in 2022 some of them are statistically significant.

9. Overall evaluation of the Japanese fight against death in 2020-22.

The manuscript sounds quite alarming. However, previous studies presented Japan as a
country with low/very moderate losses and as a “success story”.

It would be good to address this in the Discussion.

9. Language.

The manuscript would benefit from its further editing by a native English
speaker.

For example, the title “Changes in life span equity from 2020-22 in Japan” is
confusing. It would be possible to say “in 2020-22” or from “2019 to 2022”.

Reviewer #2: General Assessment

This manuscript presents a detailed demographic analysis of mortality in Japan during
the COVID-19 pandemic up to 2022, leveraging prefectural life expectancy estimates,
Arriaga decomposition, and life span equality (ℎ) derived from life table entropy.
The study updates previous work by the authors and provides novel insights on the
role of cardiovascular mortality, the discrepancies between COVID-19 statistics and
demographic indicators, and the evolution of lifespan equality.

The topic is relevant to demography, epidemiology, and public health. The methods are
standard and appropriate, the figures are generally clear, and the study addresses
an important gap: understanding indirect mortality effects and the changing age
pattern of mortality in Japan.

However, the manuscript would benefit from substantial clarification, stronger
methodological justification, better contextualization, and more cautious
interpretation. Some claims are not fully supported by the analyses provided, and
certain demographic measures are introduced without sufficient explanation for a
multidisciplinary readership.

Overall, the manuscript is publishable after major revisions.

Major Comments

Interpretation of prefectural analysis needs caution. The manuscript argues that the
absence of correlation between prefectural COVID-19 statistics and life expectancy
change in 2021–22 indicates “growing ascertainment bias”. With only 47 prefectures,
linear regression power is limited. COVID-19 cases, ICU days, and deaths suffer from
known reporting issues, but the manuscript does not quantify potential
misreporting.

Life expectancy changes include all-cause mortality, not just COVID-19: structural
health-system strain, ageing, deferred healthcare, and influenza resurgence might
all explain discrepancies. Instead of asserting ascertainment bias, acknowledge it
as one possible explanation among others, and emphasize the ecological, multi-cause
nature of the prefectural relationships.

Cause-of-death decomposition requires deeper contextualization. The decomposition
clearly shows increased contributions of cardiovascular and “other” causes in 2022
(Fig. 3) . However, “other causes” is too broad. Senility, accidents, metabolic
disorders? If senility drives most changes, this should be explicitly shown, not
only mentioned.

Cardiovascular increases are attributed to COVID-19-related indirect effects, but the
manuscript does not present evidence for causal mechanisms. Add a supplemental table
with the breakdown of “other causes.” Add a paragraph clarifying whether increases
in cardiovascular deaths follow known international post-COVID trends or reflect
Japan-specific dynamics.

Life-span equality section is underdeveloped. The discussion of ℎ (life span
equality) is technically correct but insufficient for non-specialist readers.
Interpretation is vague: “undesirable increase in life span equality” (lines
314-315) may confuse readers unfamiliar with life table entropy.

The explanation linking increased ℎ to higher mortality above threshold age a_h is
technically valid but described in overly mathematical terms. Life span equality
increased because mortality increases were concentrated at older ages, which
compresses the age-at-death distribution. Provide a short, intuitive explanation of
life table entropy and its link to lifespan compression.

The Results section sometimes repeats earlier work. Some paragraphs replicate
findings already described in the authors’ 2019–2021 study (Munira et al., 2023),
but without clearly distinguishing what is new in the update up to 2022.

Add a short subsection explicitly stating, What was already known from 2019–2021?
What has changed in 2022? Why 2022 is important for Japan (e.g., Omicron, high
vaccine coverage, healthcare saturation)?

Graphical presentation could be improved.

Fig. 4 (Life span equality). The vertical axis lacks interpretation: add labels or
annotations explaining what changes of 0.02 in ℎ represent.

Methods need more transparency. Missing clarifications. Were abridged life tables
validated against official JMD single-year tables? Authors mention near-identity but
show no numbers.

What values were used for ax in 2020–2022? JMD uses specific ax schedules that change
in crisis years. Add a short appendix or supplemental note describing the ax
assumptions, interpolation methods, quality checks.

Discussion of policy relevance could be strengthened. The manuscript ends with
general statements but does not articulate the policy implications of increased
cardiovascular mortality, increased senility deaths, decoupling of reported COVID-19
statistics from demographic indicators. A clearer final section would improve
impact.

Minor Comments

In the first sentence of the abstract, life expectancy is not wrongly defined. It
says that “Life expectancy is a demographic measure of the death structure and its
change at the population level.” But LE does not measure the death structure
(whatever that means); it is one of the central measures in demography and actuarial
science, expressing the average length of life that a newborn (or any person of a
given age) is expected to live, if current mortality conditions were to remain
constant throughout their remaining lifetime. Formally, life expectancy is a summary
measure of mortality derived from a life table, which models the survival pattern of
a hypothetical cohort subjected to age-specific mortality rates observed in a
particular population and period.

The objective of the manuscript “…the present study focuses on interpreting changes
in the nature of mortality not only by life expectancy but also by the age
distribution of mortality in Japan” is confusing by how it is written. I think the
focus goes beyond that.

Copy-editing is needed.

Neoplastics does not exist as a cause of death; perhaps you mean neoplasms.

In the Abstract says: “Beyond this measure that have been heavily applied…” must be
“…has been heavily applied.”

Clarify what life-span equality measures in one sentence.

Introduction

Provide more background on Japanese COVID-19 mortality, including excess mortality
patterns reported by other sources.

Methods

Add citations when introducing the formulas. Perhaps Preston, Heuveline and Guillot
(2001). Actually, you could discard the explanation of the life table construction;
just refer to it , for example, Preston et al. (2001).

Preston, S. H., Heuveline, P., & Guillot, M. (2001). Demography: Measuring and
modeling population processes . Blackwell Publishers.

Clarify whether decomposition was symmetric or stepwise.

Results

Avoid repeating numeric results from S1 Table in multiple places streamline. You
started the results by quoting it. If it is important, just include it in the text,
not in the appendix.

Discussion

Statements such as “not surprising” should be replaced with evidence-based
justification.

References

Some citations appear duplicated or misformatted.

If you choose “no”, your identity will remain anonymous but your review may still be
made public.

**Do you want your identity to be public for this peer review?** For
information about this choice, including consent withdrawal, please see our For
information about this choice, including consent withdrawal, please see our For
information about this choice, including consent withdrawal, please see our For
information about this choice, including consent withdrawal, please see our
Privacy Policy..

Reviewer #1: No

Reviewer #2: No

---

## [Author Response · Author response to Decision Letter 1]

9 Jan 2026

[Point-by-point responses to Reviewers] Changes in life expectancy and life span
equality during the COVID-19 epidemic in 2020-22 in Japan (Previously “Changes in
life expectancy and life span equality during the COVID-19 epidemic in Japan up to
2022.”) (submitted to PLOS One)

[Response to Reviewer 1]

We appreciate your thorough and very helpful review. Please find below the
point-by-point response to your comment.

The Introduction and Discussion sections do not mention some relevant studies on the
COVID-related mortality elevation in Japan. Namely, these are:

Hirokazu Tanaka, Shuhei Nomura, Kota Katanoda, Changes in Mortality During the
COVID-19 Pandemic in Japan: Descriptive Analysis of National Health Statistics up to
2022, Journal of Epidemiology, 2025, Volume 35, Issue 3, Pages 154-159

Tanaka H, Togawa K, Katanoda K. Impact of the COVID-19 pandemic on mortality trends
in Japan: a reversal in 2021? A descriptive analysis of national mortality data,
1995–2021. BMJ Open 2023;13:e071785. doi:10.1136/ bmjopen-2023-071785

Z Shervani,AA Khan, I Khan, A Sherwani, PDM Kumar et al. Marginal Shortening of Life
Expectancy in Japan During COVID-19: A Low Pandemic Impact Country Due to Improved
Health Infrastructure and Awareness. (2024). European Journal of Medical and Health
Sciences, 6(6), 9-17. https://doi.org/10.24018/ejmed.2024.6.6.2214

Zameer Shervani, Aamir Akbar Khan, Intazam Khan et al. Factors Explaining Japan’s Low
COVID-19 Mortality: Comparison with Rich and Democratic Countries. (2025). European
Journal of Medical and Health Sciences, 7(1), 1-9. https://doi.org/10.24018/ejmed.2025.7.1.2245

Mst Sirajum Munira, Okada Y, Nishiura H. 2023. Life-expectancy changes during the
COVID-19 pandemic from 2019–2021: estimates from Japan, a country with low pandemic
impact. PeerJ 11:e15784 DOI 10.7717/peerj.15784

These recent studies are thematically close to the present study. They should be
mentioned in the Introduction and/or Discussion.

Similarities or differences between the present study and the earlier studies should
be noted. The readers should know what novelty the present study adds to the
existing literature.

>>

Response:

We agree that these references should be referred to in the manuscript. We added the
suggested references as suggested, with relevant descriptions added to the original
manuscript. (L82-83, L85-91, L363-366, L375-L377)

The study focuses on annual changes in e0 in 2020-22. However, the full amount of e0
losses in these years can be assessed only through comparisons between the observed
e0 values and the counterfactual (predicted) e0 values. Using the Human Mortality
Database (HMD) series for Japan, one can see that from 2012 to 2019, the Japanese
life expectancy was increasing by 0.18 years per calendar year on average. If we
apply this rate of increase to 2020-22, the e0 losses for Japan in 2021 and 2022
would be 0.38 and 0.98 years instead of the observed (annual declines) of 0.13 and
0.48 years.

This difference is important and should be acknowledged. It would also be interesting
to see speculations about possible differences between the annual decreases in e0 in
2020-22 and the e0 losses in the same years.

>>

Response:

We agree that we should have added counterfactual e0 values for 2020-22. We added the
comparison of actual (observed) life expectancy and a counterfactual LE
(10-year-average change from 2010-2019 is maintained up to 2022) to the new “Table
1” in the revised manuscript (former S1 Table)

The manuscript has a massive Methods section. It is not clear whether it is a
methodology paper or a substantive analysis. There are numerous formulae. All of
them are known from earlier studies. The first seven equations are completely
unnecessary, as they are very conventional and can be found in demography textbooks,
such as those by Preston or Keyfitz, or Chiang.

>>

Response:

We agree that the original manuscript included conventional methodologies which can
be spared in the main text. We moved such descriptions to the supplement, and added
relevant references to the main text as suggested. (L 105-114, S1 Methods)

The following equations for the analysis of the age at death equality h are less
trivial since they are newer. Maybe the formal definitions of the entropy H, e+
(e-dagger), and the equality h could remain in the main text. The next four
equations related to further analysis of temporal changes in h should be given in a
supplementary appendix.

Instead, the reader needs a transparent explanation of all measures and calculation
procedures. The substantive sense of h should be clarified. Without a transparent
explanation, it's impossible to understand why h is used and what is the public
health sense of its decrease by 0.07 between 2019 and 2020.

Perhaps it would be better to use e+ instead of h. e+ is a transparent measure of
life expectancy losses by Vaupel and Canudas-Romo. e+ is counted in years of life
and has a clear public health sense (Shkolnikov et al. 2011).

Shkolnikov V.M., Andreev E.M., Zhang Z., Oeppen J., J.W.Vaupel. 2011. Losses of
expected lifetime in the United States and other developed countries: methods and
empirical analyses. Demography, 48: 211-239

>>

Response:

Equations

We moved four equations about the temporal changes in h to S4 Methods.

Explanation on measures and calculation procedures

We agree that the interpretation of h and e+ should have been clearly stated. To
address this issue, we added descriptions

-to that help readers understand e+(reference added), and the difference between e0,
e+, h, H (L132-138)

-to help interpret changes in H or h = -log(H) (L120~ L130)

Choice of metric to evaluate life span equality or life disparity

We clarified the merit of evaluating h in L139-145, with an intuitive example to
guide interpretation. We also agree that e+ is a measure that should be highlighted
in the main analysis. Therefore, in the result section we also

-changed panel (B) in Fig 4 to a 2D plot showing the historical relationship between
∆ log⁡(e_0 ) and ∆ log⁡(e^† ), so that what we described in L139-145 links to the
results.

-added S10-S12 Fig that shows the decomposition results of the yearly changes of e+
by age and cause from 2019-20 to 2021-22, which helps understand the relationship
between h and e+, with relevant descriptions in the result section. (L288-L291)

Despite the length of the Methods, they lack a precise description of the data and
computational procedures. Which calculations were based on the complete life tables
vs. the abridged life tables? It is clearly preferable to use the single-year-age
data for calculating dispersion measures such as h or e+. This information is
needed. It can be provided in a supplementary appendix.

>>

Response:

We should have provided precise information on this issue.

For the calculation of h and e+, we used the complete life tables provided by JMD.
For prefectural analysis and Arriaga decomposition, we used the abridged life tables
we re-calculated based on the abridged life tables provided in JMD

We added relevant explanation. (L105- L114, L130)

The analysis of causes of the h changes between 2019 and 2020, 2020 and 2021, 2021
and 2022 is not convincing. From the methodology point of view, it is possible to
see directly the components of changes in h. This could be done by decomposing the
change in h by age and CODs with one of the universal decomposition methods
(discrete replacement method by Andreev et al. 2002; continuous change method by
Horiuchi et al. 2008). These methods estimate additive contributions of different
ages and causes of death to a change in any aggregate measure. This has not been
done. Looking at changes in M(x) or at the weights W(x) cannot replace the direct
decomposition.

On the substantive side, there are also some problems. First, the alarming drop in h
between 2019 and 2020 is not explained. However, Figure 3 reveals moderate mortality
increases at young adult ages in 2020. My own quick look at the Human Mortality
Database data for Japan also indicates mortality increases at ages between 20 and 29
as well as some ages between 30 and 44 in 2020. So, what happened in 2020? It is
important to know. The potential role of the pandemic is doubtful. Second, increases
in h from 2020 to 2021 and from 2021 to 2022 are considered “undesirable” as they
were (likely) caused by the mortality elevation at old and very old ages. At the
same time, the h increase in 2021-22 may be considered an advantage of Japan
compared to other countries. While in Japan, the mortality increase in 2021 and
especially in 2022 was entirely concentrated at old ages, in many other countries
(USA, Eastern Europe, Britain, Belgium, and other), the COVID-related mortality
increases were also large at midlife and “young old” ages (55-69). The latter
pattern results in stagnant or even decreasing values of h.

That is to say that the changes in e0 and e+ in Japan should be considered in context
of e0 and e+ changes elsewhere.

>>

Response:

Thank you very much for your very important suggestion.

Decomposition of h by age and cause of deaths

We conducted decomposition by applying the approach of Horiuchi et al as suggested.
The relevant methodologies were added in the methods section (L138-L146, S3
Methods), and the results are presented in Fig 5 and in S1 Data. (We also provided
the same decomposition results for e+ in S10-S12 Fig.)

Mortality increase in young adults in 2020:

We should have provided more comprehensive explanation for this. Suicide as an
important cause of death that elevated mortality in 2020 among young adults in Japan
has been revealed in published studies. Together with these preceding studies, we
also added a supplementary analysis (S6 Table) to clarify the impact of suicide in
those aged 10-44 in 2020. This is mentioned in the relevant part of the discussion
section. (L363-368)

Further interpretation of “h” from 2020-22.

We agree that “increase in h” may not necessarily be an “undesirable” consequence as
we argued in the original text, without further analyses.

To address this issue, firstly, we added a new figure panel in Fig 4 that compares ∆
log⁡(e_0 ) against ∆ log⁡(e^† ).

This revealed that, by visual inspection, 2019-20 was an outlier in that ∆ log⁡(e^† )
increased substantially while positive ∆ log⁡(e_0 ) was maintained. However, after
2019-20, it turned out that both ∆ log⁡(e_0 ) and ∆ log⁡(e^† ) turned negative and
grew larger in magnitude up to 2021-22.

We also placed Fig 5 that presents the results of decomposition of ∆h by age and
cause in 2019-20, 2020-21, 2021-22, which now clearly describes the contribution of
elderly deaths as the leading cause of “increase in h” in 2020-21 and 2021-22,
whereas contributions from younger adults were overall negative to neutral
overall.

Discussion about the large increases in mortality from CVD in 2021 and 2022 does not
take into account potential direct contributions of COVID-19 to deaths from CVD.
COVID-19 could appear on the medical death certificate as an immediate,
intermediate, or contributory cause. It would be good to discuss how large the
direct influence of COVID-19 could be in CVD deaths in Japan.

>>

Response:

We regret that this issue should rather be discussed as a limitation of our study,
because there is no public data to explore the possible bias regarding the death
certificate. However, we may argue the potential effect of COVID-19 on CVD, given
that there is a published study on age-standardized mortality rates by cause of
death in Japan suggests an increase in ASMR due to “Heart diseases” that is roughly
proportional to increase in ASMR due to “COVID-19”. We added relevant discussions in
the revised manuscript (L341-L348)

The analysis of statistical associations between e0 decreases in 2021 and 2022 and
the epidemiological indicators is not fully justified. The authors should perform
statistical tests for normality, heteroscedasticity, and linearity. Pearson’s r
values together with p-values should be shown on these graphs. The regression
outcomes are commented as if there is a radical difference between 2021 and 2022. It
looks like statistical links exist in 2021 and they do not exist in 2022. However,
the scatterplots and the tables show that in both years, the links are very weak,
but in 2022 some of them are statistically significant.

>>

Response:

We agree we should have provided richer statistical tests and consideration for
normality, heteroscedasticity, and linearity (L 119-128)

Firstly, we added Pearson’s r values with p-values as suggested to the plots.

In our linear regression including only the 1st order (linear) term, Shapiro-Wilk
test suggested deviation from normality in residuals for the “2022 person-days in
intensive care “ OLS. The Breusch-Pagan test also suggested heteroscedasticity in
“2021 person-days in intensive care” and “2021 deaths”. (S1 Table)

To account for heteroscedasticity, we report inference based on
heteroscedasticity-robust (sandwich) standard errors throughout.

To assess potential non-linearity, we conducted comparison between linear regression
models with “1st order term (assuming linearity; main analysis)” with “1st + 2nd
(quadratic) order terms (assuming non-linearity) in this framework. Across all
analyses, the quadratic term did not improve model fit and the Wald test did not
support choosing quadratic models. Thus, our main analyses are based on linear
specification.(S3 Table)

Given that non-normality was suggested in some models, we also conducted wild
bootstrap inference to obtain robust standard errors for validating the results from
sandwich estimators. The results from wild bootstrap were consistent with those from
sandwich estimators in all analyses. (S2 Table)

To visualize the “slope change” from 2020-21 to 2021-22, we also conducted
supplementary analyses by the following regression model: (in S2 Method)

∆e_0~ β_0+β_1 1_(2021-22)+(β_2+β_3 1_(2021-22) ) log⁡(x),

Which essentially yielded same results as in individual anaiyses but provides more
clarity. These supplementary analyses revealed non-significant change in the
“slopes” in all models from 2020-21 to 2021-22. Based on these results, we also
revised the text to avoid overstating year-to-year differences between 2021 and
2022.(S4 Table)

Overall evaluation of the Japanese fight against death in 2020-22: The manuscript
sounds quite alarming. However, previous studies presented Japan as a country with
low/very moderate losses and as a “success story”. It would be good to address this
in the Discussion.

>>

Response:

We added relevant discussion in L315-323 with additional references as suggested.

Language. The manuscript would benefit from its further editing by a native English
speaker. For example, the title “Changes in life span equity from 2020-22 in Japan”
is confusing. It would be possible to say “in 2020-22” or from “2019 to 2022”.

>>

Response:

We revised the title as suggest. We also checked our English writing again throughout
the revised manuscript.

[Response to Reviewer 2]

We truly appreciate your very fundamental advice on our manuscript. Please find below
the point-by-point response to your comment.

<Major>

Interpretation of prefectural analysis needs caution. The manuscript argues that the
absence of correlation between prefectural COVID-19 statistics and life expectancy
change in 2021–22 indicates “growing ascertainment bias”. With only 47 prefectures,
linear regression power is limited. COVID-19 cases, ICU days, and deaths suffer from
known reporting issues, but the manuscript does not quantify potential
misreporting.

>>

Response:

We agree that our results only suggest correlation and the message should be about
suggestions on one possible mechanism that may underly our findings. We set a lower
ton on this argument in relevant sentences (L198-206, L333-338)

In addition to correcting the overstatement of our findings here, we upgraded our
analysis using linear regressions as described in L119- L128 and S1-S4 Table.

Life expectancy changes include all-cause mortality, not just COVID-19: structural
health-system strain, ageing, deferred healthcare, and influenza resurgence might
all explain discrepanci

---

## [Decision Letter · Decision Letter 1]

3 Mar 2026

Dear Dr. Nishiura,

Thank you for submitting your manuscript to PLOS ONE. After careful consideration, we
feel that it has merit but does not fully meet PLOS ONE’s publication criteria as it
currently stands. Therefore, we invite you to submit a revised version of the
manuscript that addresses the points raised during the review process.

Please submit your revised manuscript by Apr 17 2026 11:59PM. If you will need more
time than this to complete your revisions, please reply to this message or contact
the journal office at plosone@plos.org. When
you're ready to submit your revision, log on to https://www.editorialmanager.com/pone/ and select the 'Submissions
Needing Revision' folder to locate your manuscript file.. When you're ready to
submit your revision, log on to https://www.editorialmanager.com/pone/ and select the 'Submissions
Needing Revision' folder to locate your manuscript file.. When you're ready to
submit your revision, log on to https://www.editorialmanager.com/pone/ and select the 'Submissions
Needing Revision' folder to locate your manuscript file.. When you're ready to
submit your revision, log on to https://www.editorialmanager.com/pone/ and select the 'Submissions
Needing Revision' folder to locate your manuscript file.

We look forward to receiving your revised manuscript.

Kind regards,

Claudio Alberto Dávila-Cervantes, Ph.D.

Academic Editor

PLOS One

Journal Requirements:

Reviewers' comments:

Reviewer's Responses to Questions

**Comments to the Author**

Reviewer #1: All comments have been addressed

Reviewer #2: All comments have been addressed

2. Is the manuscript technically sound, and do the data support the
conclusions?

Reviewer #1: Yes

Reviewer #2: Yes

3. Has the statistical analysis been performed appropriately and rigorously?


Reviewer #1: Yes

Reviewer #2: Yes

4. Have the authors made all data underlying the findings in their manuscript fully
available?

Reviewer #1: Yes

Reviewer #2: Yes

5. Is the manuscript presented in an intelligible fashion and written in standard
English?

Reviewer #1: Yes

Reviewer #2: Yes

Reviewer #1: After the revision, the manuscript looks much better. All my comments
were taken into account and addressed through changes to the text and/or additional
analyses (decomposition of the h change and others). More of the earlier literature
is cited; the Methods section was shortened and is free of trivial/textbook
equations; the counterfactual approach was added to avoid underestimating life
expectancy losses; and the analysis of statistical associations across prefectures
is described more comprehensively. Importantly, it is mentioned now that COVID-19
may underlie some of the increases in deaths from CVD, senility, and some other
causes.

This is all good. Nevertheless, I have noticed a few minor problems to be corrected
before publication of the study.

L63-L64. “…measurable mortality impact…” – better to say “considerable” or
“substantial” impact.

L85-L86. “We also decomposed the year-on-year life expectancy change from 2019-22 …”
sounds confusing. It should be “We also decomposed annual life expectancy changes in
2019-2022.”

L101. “The deaths counts by cause of death in Japan is only available…” It should be
“The death counts by cause of death in Japan are only available…”

L135. “from 2000-2022” It should be “from 2000 to 2022” or “for the period
2000-2022”

L141-L143. It is still unclear why it is important (from the public health viewpoint)
to know whether a relative e0 change is larger or smaller than the relative e-dagger
change.

L225-L226. It is better to say “Life expectancy changes from 2020 to 2021 and from
2021 to 2022”. The column heading “Year” is confusing. It is better to say
“Period”.

L233-L235. “… more eminent in 2021-22 than in 2020-21”. This is unclear. It would be
better to say here “more eminent in the change from 2021 to 2022 than in the change
from 2020 to 2021”.

L328. “The unclear correlation …. indicators of COVID-19…”. It is better to say “The
unclear correlations across prefectures … indicators of COVID-19 …”.

Reviewer #2: (No Response)

If you choose “no”, your identity will remain anonymous but your review may still be
made public.

**Do you want your identity to be public for this peer review?** For
information about this choice, including consent withdrawal, please see our For
information about this choice, including consent withdrawal, please see our For
information about this choice, including consent withdrawal, please see our For
information about this choice, including consent withdrawal, please see our
Privacy Policy..

Reviewer #1: **Yes:** Vladimir M. ShkolnikovVladimir M. ShkolnikovVladimir
M. ShkolnikovVladimir M. Shkolnikov

Reviewer #2: No

---

## [Author Response · Author response to Decision Letter 2]

4 Mar 2026

[Point-by-point responses to Reviewers] Changes in life expectancy and life span
equality during the COVID-19 epidemic in 2020-22 in Japan (Previously “Changes in
life expectancy and life span equality during the COVID-19 epidemic in Japan up to
2022.”) (submitted to PLOS One)

[Response to Reviewer 1]

We truly appreciate your review to improve our manuscript. Please find below the
point-by-point response to your comment.

1) L63-L64. “…measurable mortality impact…” – better to say “considerable” or
“substantial” impact.

>>

Response: We changed the expression as advised.

(L61)

2) L85-L86. “We also decomposed the year-on-year life expectancy change from 2019-22
…” sounds confusing. It should be “We also decomposed annual life expectancy changes
in 2019-2022.”.

>>

Response: We changed to decomposed annual life expectancy changes” for clarity.

(L83)

3) L101. “The deaths counts by cause of death in Japan is only available…” It should
be “The death counts by cause of death in Japan are only available…”

>>

Response: We corrected this sentence accordingly.

(L98)

4) L135. “from 2000-2022” It should be “from 2000 to 2022” or “for the period
2000-2022”

>>

Response: We corrected this sentence.

(L130)

5) L141-L143. It is still unclear why it is important (from the public health
viewpoint) to know whether a relative e0 change is larger or smaller than the
relative e-dagger change.

>>

Response: We added descriptions to clarify the importance of evaluating the changes
of e-dagger in comparison to changes of e0. (L136-141)

6) L225-L226. It is better to say “Life expectancy changes from 2020 to 2021 and from
2021 to 2022”. The column heading “Year” is confusing. It is better to say
“Period”.

>>

Response: We modified the title of Table 2 and the column heading as suggested.

(L220-222)

7) L233-L235. “… more eminent in 2021-22 than in 2020-21”. This is unclear. It would
be better to say here “more eminent in the change from 2021 to 2022 than in the
change from 2020 to 2021”.

>>

Response: We revised the text as suggested.

(L228-229)

8) L328. “The unclear correlation …. indicators of COVID-19…”. It is better to say
“The unclear correlations across prefectures … indicators of COVID-19 …”.

>>

Response: We added “across prefectures” in the relevant sentence.

(L320)

---

## [Editor Report · Decision Letter 2]

9 Mar 2026

Changes in life expectancy and life span equality during the COVID-19 epidemic in
2020-22 in Japan

PONE-D-25-54870R2

Dear Dr. Nishiura,

We’re pleased to inform you that your manuscript has been judged scientifically
suitable for publication and will be formally accepted for publication once it meets
all outstanding technical requirements.

Kind regards,

Claudio Alberto Dávila-Cervantes, Ph.D.

Academic Editor

PLOS One
---

## [Editor Report · Acceptance letter]

PONE-D-25-54870R2

PLOS One

Dear Dr. Nishiura,

I'm pleased to inform you that your manuscript has been deemed suitable for
publication in PLOS One. Congratulations! Your manuscript is now being handed over
to our production team.

Kind regards,

on behalf of

Mr. Claudio Alberto Dávila-Cervantes

Academic Editor

PLOS One